# Detection of Transgenes in Gene Delivery Model Mice by Adenoviral Vector Using ddPCR

**DOI:** 10.3390/genes10060436

**Published:** 2019-06-08

**Authors:** Takehito Sugasawa, Kai Aoki, Koichi Watanabe, Koki Yanazawa, Tohru Natsume, Tohru Takemasa, Kaori Yamaguchi, Yoshinori Takeuchi, Yuichi Aita, Naoya Yahagi, Yasuko Yoshida, Katsuyuki Tokinoya, Nanami Sekine, Kaoru Takeuchi, Haruna Ueda, Yasushi Kawakami, Satoshi Shimizu, Kazuhiro Takekoshi

**Affiliations:** 1Laboratory of Laboratory/Sports medicine, Division of Clinical Medicine, Faculty of Medicine, University of Tsukuba, 1-1-1 Tennodai, Ibaraki 305-8577, Japan; take0716@krf.biglobe.ne.jp (T.S.); y-kawa@md.tsukuba.ac.jp (Y.K.); 2Graduate School of Comprehensive Human Sciences, University of Tsukuba, Tsukuba 305-8577, Japan; fineday0126@gmail.com (K.A.); s1921312@s.tsukuba.ac.jp (K.Y.); ilbsk5@yahoo.co.jp (K.T.); s1721298@s.tsukuba.ac.jp (N.S.); s1821255@s.tsukuba.ac.jp (H.U.); 3Faculty of Health and Sport Sciences, University of Tsukuba, Tsukuba 305-8577, Japan; watanabe.koichi.ga@u.tsukuba.ac.jp (K.W.); takemasa.tohru.gm@u.tsukuba.ac.jp (T.T.); yamaguchi.kaori.gf@u.tsukuba.ac.jp (K.Y.); shimizu.satoshi.fe@u.tsukuba.ac.jp (S.S.); 4Molecular Profiling Research Center for Drug Discovery, National Institute of Advanced Industrial Science and Technology (AIST), Tokyo 100-8921, Japan; t-natsume@aist.go.jp; 5Nutrigenomics Research Group, Faculty of Medicine, University of Tsukuba, Ibaraki 305-8575, Japan; yoshinori-takeuchi@umin.ac.jp (Y.T.); metabmetabmetab@gmail.com (Y.A.); nyahagi-tky@umin.ac.jp (N.Y.); 6Department of Medical Technology, Faculty of Health Sciences, Tsukuba International University, 6-20-1 Manabe, Tsuchiura, Ibaraki 300-0051, Japan; shinbelon@mwa.biglobe.ne.jp; 7Japan Society for the Promotion of Science; Kojimachi Business Center Building, Kojimachi, Chiyoda-ku, Tokyo 102-0083, Japan; 8Laboratory of Environmental Microbiology, Division of Basic Medicine, Faculty of Medicine, University of Tsukuba, Tsukuba 305-8575, Japan; ktakeuch@md.tsukuba.ac.jp

**Keywords:** gene doping, gene therapy, droplet digital PCR, adenoviral vector

## Abstract

With the rapid progress of genetic engineering and gene therapy, the World Anti-Doping Agency has been alerted to gene doping and prohibited its use in sports. However, there is no standard method available yet for the detection of transgenes delivered by recombinant adenoviral (rAdV) vectors. Here, we aim to develop a detection method for transgenes delivered by rAdV vectors in a mouse model that mimics gene doping. These rAdV vectors containing the *mCherry* gene was delivered in mice through intravenous injection or local muscular injection. After five days, stool and whole blood samples were collected, and total DNA was extracted. As additional experiments, whole blood was also collected from the mouse tail tip until 15 days from injection of the rAdv vector. Transgene fragments from different DNA samples were analyzed using semi-quantitative PCR (sqPCR), quantitative PCR (qPCR), and droplet digital PCR (ddPCR). In the results, transgene fragments could be directly detected from blood cell fraction DNA, plasma cell-free DNA, and stool DNA by qPCR and ddPCR, depending on specimen type and injection methods. We observed that a combination of blood cell fraction DNA and ddPCR was more sensitive than other combinations used in this model. These results could accelerate the development of detection methods for gene doping.

## 1. Introduction

Doping is an act of raising competitive abilities to achieve success by using substances or methods prohibited in sports [1]. Doping in sports, especially in festivals such as the Olympic Games and in world or local championships for various competitions, is considered illegal and against the spirit of the game. The World Anti-Doping Agency (WADA) was established in 1999, and has been involved in scientific research on doping, anti-doping education, development of anti-doping strategies, and monitoring of the World Anti-Doping Code (hereafter the Code) [2] to ensure soundness and fairness in sports worldwide.

With the rapid progress of genetic engineering technology and gene therapy, WADA has been strongly alerted against gene doping. Since its early days, WADA has added “gene doping” to its prohibited list. Subsequently, in 2004, WADA created a panel of experts on gene doping to investigate the latest advances in the field of gene therapy, and the methods for detecting doping [3]. In January 2018, WADA extended the ban on gene doping to include all forms of gene editing. Therefore, the list of prohibited substances currently includes “gene editing agents designed to alter genome sequences and/or the transcriptional or epigenetic regulation of gene expression” [4]. However, there are no established standard methods for detecting or preventing gene doping to date.

In recent years, genetic engineering technology has rapidly advanced, resulting in the progression of gene therapy. In gene therapy, various viral vectors have been frequently devised and applied. Vectors based on recombinant adeno-associated viruses (rAAV) and recombinant adenoviruses (rAdV) have been widely used in clinical trials and animal experiments for investigating gene therapy. For example, rAAV vectors have been applied in the treatment of diseases, such as Duchenne muscular dystrophy (DMD) [5], hemophilia B [6,7], and Leber congenital amaurosis (LCA) [8,9], during clinical trials or animal experiments as a form of gene therapy. Moreover, rAdV vectors also have been applied in gene therapy for the treatment of certain human cancers [10,11,12]. In China, two rAdV vector-based gene therapy products, namely Gendicine (Shenzhen SiBiono GeneTech Co., Ltd., Shenzhen, China) [12,13] and Oncorine (Sunway Biotech Co., Ltd., Taipei, Taiwan), were approved for clinical use in humans to treat head and neck cancer, and were released into the commercial market in 2003 and 2006, respectively [14]. Additionally, rAdV vectors were the most commonly used vectors in approved clinical trials of gene therapy (541 cases, 18% of the total) worldwide until December 2018 [15] (Table 1). It can be assumed that gene doping methods may employ clinical trial methods. Therefore, there is a possibility that rAAV or rAdV vectors, especially rAdV vectors, can be used as gene doping agents to enhance athletic performance by artificially modifying gene expression in specific human organs. In this study, we focused on rAdV vectors, since rAdV vectors are the most commonly used in clinical trials [15] (Table 1), and are also used as prescription drugs [12,13,14].

The data in Table 1 were obtained from the website of Gene Therapy Clinical Trials Worldwide [15]. The top seven vectors include five viral vectors, with rAdV vectors being the most commonly used. It is believed that these viral vectors can be used for transgene delivery in gene doping.

## 2. Materials and Methods

### 2.1. Cloning of Recombinant Adenoviral Vectors Containing the mCherry Gene

The following plasmids were used in this study: plasmid pcDNA3.1–Peredox–*mCherry* was a gift from Gary Yellen [16] (Addgene plasmid #32383; http://n2t.net/addgene:32383; RRID: Addgene_32383); *pENTR4* (Thermo Fisher Scientific, Waltham, MA, USA); and *pAd*/*CMV*/*V5-DEST* (Thermo Fisher Scientific). HEK 293A cells (Thermo Fisher Scientific) were used to clone and amplify recombinant adenoviral (rAdV) vectors. The *mCherry* gene, having restriction enzyme sites of 5′-EcoRI and 3′-NotI, was amplified by PCR with templated *pcDNA3.1*–*Peredox*–*mCherry*. It was then cloned into a *pENTR4* plasmid between EcoRI and NotI sites by restriction enzyme digestion, followed by ligation with T4 ligase (Promega, Madison, WI, USA). The sequences of inserted *mCherry* genes in the pENTR4 plasmids were read using sanger sequencing and confirmed to be correct sequences. Using Gateway LR Clonase Enzyme mix (Thermo Fisher Scientific), and according to the manufacturer protocol, *pENTR4* containing the *mCherry* gene was allowed to react and recombine with *pAd*/*CMV*/*V5-DEST* (destination vector) in an LR reaction to move the *mCherry* gene into a *pAd*/*CMV*/*V5*-*DEST* plasmid, which can make rAdV type 5, containing the transgene. Subsequently, a *pAd*/*CMV*/*V5-DEST* plasmid containing the *mCherry* gene was digested with Pac I restriction enzymes (New England Biolabs, Ipswich, MA, USA), and the resulting liner plasmids were transfected using Lipofectamine LTX Reagent (Thermo Fisher Scientific) into HEK 293A cells cultured in Dulbecco’s Modified Eagle Medium (DMEM; Thermo Fisher Scientific), containing 10% Fetal Clone III (GE Healthcare, Chicago, IL, USA) and antibiotics (Nacalai tesque, Kyoto, Japan) to synthesize and amplify rAdV vectors containing the *mCherry* gene. Amplified rAdV vectors were purified by CsCl density gradient ultracentrifugation followed by gel filtration, according to the protocol described by Takeuchi et al. [17,18]. The concentration of rAdV viral particles (VP) was measured on a spectrophotometer, according to the method of Sweeney and Hennessey [19]. To confirm the expression of a functional mCherry protein, HEK 293A cells were seeded at a density of 2.5 × 10^5^ cells per well in six-well plates, and were cultured in DMEM containing 10% Fetal Clone III and antibiotics. After 24 h, the cells were infected with rAdV vectors (2.8 × 10^9^ VP/mL of medium) to allow the expression of *mCherry*. After 24 h of induction, the red fluorescence of mCherry was analyzed using fluorescence microscope.

### 2.2. Animal Experiments

Animal experiments in this study were approved by the Animal Care Committee, University of Tsukuba (approval numbers: 18–118 and 18–474). The overview of the experiments is shown in Figure 1. Six-week-old IC57BL/6 male mice were purchased from the Central Laboratories for Experimental Animals (CLEA, Tokyo, Japan). The mice were allowed to grow until they reached 10 weeks old, with an average body weight of 26.1 g (SD = ±1.8 g). At this point, they were sacrificed for further experiments.

#### 2.2.1. Acute Experiments

The rAdV vectors containing the *mCherry* gene (2.1× 10^11^ VP) were injected into left orbital veins (intravenous; IV group, *n* = 7) or local muscle (LM group, *n* = 7) of both the tibialis anterior (TA) muscles of mice under general anesthesia by inhalation agent isoflurane. When rAdV vectors were used intravenously (IV), most of the rAdV transgenes accumulated in the liver. Control mice were left untreated (Con. Group, *n* = 6). After five days of injection, mice were placed in an empty cage and were allowed to defecate. Stool samples were quickly collected into microtubes and placed on ice. After collecting stool samples from experimental mice, whole blood was extracted from the inferior vena cava using ethylenediaminetetraacetic acid (EDTA) as an anticoagulant. During this procedure, mice were given general anesthesia by inhalation agent isoflurane. After blood collection, the mice were euthanized. Whole blood was then centrifuged and separated into plasma and blood cell fraction. Liver and TA muscle were also harvested to check gene and protein expression after infection with rAdV vectors. Collected stool samples, plasma samples, and blood cell fractions were stored at −20 °C, whereas liver and TA muscle samples were stored at −80 °C till further analysis.

#### 2.2.2. Chronic Experiments

Initially, as pre-samples, 50–100 µL of whole blood was collected from mice tail tip cuttings of 2–3 mm, under general anesthesia by isoflurane. Then, mice were injected with rAdV vectors of the same amounts and by same method as those described in Section 2.2.1. After 24 h of injection, whole blood was again collected for the next 15 days, a total of eight times on every other day by the same methods. The collected whole blood was stored at −20 °C until further analysis.

### 2.3. Confirmation of Gene and Protein Expression by Infection of rAdV Vectors In Vivo

For acute experiments, the total RNA and proteins were extracted from isolated liver and TA muscle tissues as follows. Sepasol-RNA I Super G (Nacalai Tesque, Kyoto, Japan) was used for total RNA extraction, according to the manufacturer’s instructions. Using 500 ng of total RNA and PrimeScrip RT Master Mix (Takara Bio, Shiga, Japan), reverse transcription was carried out to synthesize cDNA. The cDNA synthesized was diluted 10-fold using nuclease-free water. Subsequently, quantitative PCR (qPCR) was performed to confirm *mCherry* expression in different tissues with duplicate measurements, using a KAPA SYBR Fast qPCR kit (NIPPON Genetics, Tokyo, Japan) for 18S ribosomal RNA and PrimeTime Gene Expression Master Mix (Integrated DNA Technologies) for *mCherry*, normalized to 18S ribosomal RNA expression with delta CT calculations. Primer sequences are given in Table 2. To extract the total protein, the tissues were homogenized in a lysis buffer (0 mM Tris-HCl (pH 7.4), 150 mM NaCl, 1% NP40, 1 mM EDTA) with an added protease inhibitor cocktail (Nacalai Tesque), and subjected to Western blotting using 10 μg protein sample. After the protein transfer, the membrane was incubated in TBS-T buffer (50 mM Tris-HCl, 150 mM NaCl, 0.05% Tween 20) containing 5% Bovine Serum Albumin (BSA) for blocking, and subsequently in anti-mCherry antibody (PM005; MBL) overnight at 4 °C with gentle shaking. The next day, after thorough washes, the membrane was incubated with a secondary antibody conjugated with horseradish peroxidase for 1 h with gentle shaking. Finally, the protein bands of mCherry were visualized with ECL Select Western blotting Detection Reagent (GE Healthcare) using LAS-4000 software (GE Healthcare). Glyceraldehyde-3-phosphate dehydrogenase (GAPDH) was used as a loading control (C65); (Santa Cruz Biotechnology, Sant Cruz, CA, USA).

### 2.4. DNA Extraction and Detection of Transgenes Using Three Different PCR Methods

For acute experiments, total DNA was extracted from collected stool, plasma, and blood cell fractions. A NucleoSpin Plasma XS (Takara Bio, Kusastu, Japan) kit was used to isolate plasma cell-free DNA (cfDNA) from 240 µL of plasma. A NucleoSpin Blood (Takara Bio) kit was used to isolate DNA from 200 µL of blood cell fraction. A phenol/chloroform/isoamyl alcohol solution (Nacalai Tesque) was used to isolate DNA from the stool of one piece. The concentration of total extracted DNA was measured, and the final concentration was adjusted to 50 ng/µL for stool- and blood cell fraction DNA. Since plasma cfDNA had a very low concentration, it was used as an undiluted solution. For chronic experiments, the total DNA was extracted from 50–100 µL of whole blood using NucleoSpin Blood, and its final concentration was adjusted to 50 ng/µL.

Using DNA samples of acute experiments, semi-quantitative PCR (sqPCR), real-time quantitative PCR (qPCR), and droplet-digital PCR (ddPCR) were performed to detect transgene fragments. For DNA samples of chronic experiments, only ddPCR was performed. All primer sequences used in these PCR methods are given in Table 2.

### 2.5. Semi-Quantitative PCR (sqPCR)

Kod Plus (TOYOBO, Osaka, Japan) reagent was used to perform sqPCR. The template volume and primer concentrations were 1 µL and 300 nM, respectively, for a total reaction volume of 10 µL per sample. The conditions maintained in the thermal cycler were 94 °C for 2 min; 98 °C for 10 s, 60 °C for 30 s and 68 °C for 30 s, for 35 cycles; and then 4 °C for infinite hold. The amplicons were subjected to electrophoresis and visualized using ethidium bromide in an LAS-4000 transilluminator (GE Healthcare).

### 2.6. Real-Time Quantitative PCR (qPCR)

PrimeTime Gene Expression Master Mix (Integrated DNA Technologies, Coralville, IA, USA) reagent was used to perform qPCR. The template volume, primer, and probe concentrations were 2 µL, 500 nM, and 250 nM, respectively, for a total reaction volume of 10 µL per sample. The *pcDNA3.1*–*Peredox*–*mCherry* plasmids (10 pg/µL) were used as standard DNA to perform absolute quantification. The conditions maintained in the thermal cycler were 95 °C for 3 min; and 95 °C for 3 s and 60 °C for 30 s, for 35 cycles. Melting curve was analyzed on QuantStudio 5 Real-Time PCR Systems (Thermo Fisher Scientific). All samples were measured in duplicate, and the coefficient of determination (R2) of the standard curve was equal to 0.98.

### 2.7. Droplet Digital PCR (ddPCR)

To form droplets, ddPCR Supermix for Probes and Droplet Generator oil (Bio Rad, Hercules, CA, USA) were used. The template volume, primer, and probe concentration were 1 µL, 500 nM, and 250 nM, respectively, for a total reaction volume of 20 µL per sample. Droplets were formed by an automated droplet generator (Bio Rad). The conditions maintained in the thermal cycler were 95 °C for 10 min; 94 °C for 30 s and 60 °C for 1 min, for 40 cycles; 4 °C for 5 min; 90 °C for 5 min; and 4 °C -infinite hold. The droplets were analyzed as PCR-positive or PCR-negative by QX200 Droplet Digital PCR System (Bio Rad). DNA samples from chronic experiments were also subjected to ddPCR using a similar method. All samples were measured in duplicate.

### 2.8. Statistics

The bar graph shows average ± SD and plots of individual values. The tables show individual absolute values and the median. The bar graph and table data were subjected to a Kruskal–Wallis H test (one-way ANOVA of ranks), followed by a two-stage Benjamini, Krieger, and Yekutieli False Discovery Rate (FDR) procedure as a post-hoc test, using GraphPad Prism version 7.04. A *p* value less than 0.05 was considered to be statistically significant. 

## 3. Results

### 3.1. The mCherry Gene and Protein Were Sufficiently Expressed Both In Vitro and In Vivo

The red fluorescent signal of mCherry protein after infection with rAdV vectors was confirmed in HEK 293A cells (Figure 2A). In acute experiments, RNA and protein expression of mCherry were also confirmed in the liver and TA muscle in mice infected with rAdV (Figure 2B,C).

### 3.2. The Three PCR Methods Showed Each Characteristic and Could Detect Transgene Fragments in Acute Experiments

In sqPCR, transgene fragments were detected with strong signals for blood cell fraction DNA, as well as very weak signals for other DNA samples that could not be clearly distinguished into positive or negative signals (Figure 3).

In the qPCR, using the TaqMan probe, transgene fragments were detected in all specimens in the IV group, with strong signals from the blood cell fraction DNA. However, small amounts of transgene fragments were observed in the blood cell fraction DNA, while no fragments were detected in the plasma and stool DNA of the LM group. Additionally, variations between individual values were very large (Table 3).

For ddPCR, transgene fragments were detected in all specimens in the IV group, with strong signals for DNA from the blood cell fraction. However, in the LM group, small amounts of transgene fragments were observed in the blood cell fraction DNA, and significantly lower amounts were found in stool DNA; fragments were not detected in plasma cfDNA and stool DNA. Additionally, variations between individual values were also very large (Figure 4, Table 4).

### 3.3. ddPCR on Chronic Experiments Showed a Possibility of Detecting Transgenes Repeatedly

From the results of the acute experiments, a combination method of ddPCR and blood cell fraction DNA had the highest sensitivity for detecting transgene fragments, and it was hypothesized that performing ddPCR with whole blood DNA is useful for detecting transgene fragments over several days. Therefore, we performed the combination method in the chronic experiment. In the results, transgene fragments could be detected. The transgene fragments mainly existed between one and three days, especially in the IV group, but decreased sharply after three days. The fragments could be detected for approximately seven days in the IV group, and for five days in the LM group (Figure 5, Table 5). Additionally, variations between individual values were very large.

Transgene fragments could be detected consistently after one and three days post-transfection, especially in the IV group, but decreased sharply after three days with elapsed time. IV means intravenous injection, and LM means local muscular injection of the rAdV in tibialis anterior (TA) muscle.

## 4. Discussion

We tried to develop a detection method for transgenes delivered by rAdV vectors in a gene delivery mouse model. We successfully detected transgenes containing the *mCherry* gene in blood fraction DNA, plasma cfDNA, stool DNA, and whole blood DNA, using sqPCR, qPCR, and ddPCR. Both IV and LM groups showed gene expression and functional protein expression in liver tissue and TA muscles. Therefore, our model mimics the gene delivery model for liver or local muscle, which can potentially be used in gene doping. However, the model is not a true representative of gene doping, since the aim was not to enhance muscle endurance or power of mice. Nevertheless, this model resembles gene doping in terms of rAdV vectors, since a functional protein was successfully expressed in the liver or local muscle tissue. Hence, it is believed that this model can be useful for developing novel detection methods for transgene fragments in gene doping.

After comparing combinations of three different PCR methods and three different specimens, it was observed that a combination of ddPCR and blood cell fraction DNA was highly sensitive for the detection of a transgene. However, this method has the limitation of being invasive. Moreover, we detected transgene fragments in stool DNA samples as well, although in low amounts. Therefore, there is the possibility that non-invasive sampling for gene doping could be achieved by using stool DNA.

We observed that transgene fragments predominantly exist in blood fraction. Therefore, it is recommended to use blood cell fraction for the examination of gene doping by rAdV vectors. Additionally, it is also considered that rAdV vectors are captured by neutrophils, monocytes, or macrophages, or are attached to red blood cells. To solve these questions, further experiments, such as sorting different types of cells using flow cytometry, are needed.

We hypothesized that the concept of liquid biopsy for the diagnosis or monitoring of cancer—that is, to analyze specific DNA fragments in cancer cells using plasma cfDNA—would be useful for the detection of transgene fragments. Therefore, initially in this study, we tried to detect the transgene fragment using plasma cfDNA in acute experiments, using the concept of liquid biopsy. However, the number of fragments detected in the plasma cfDNA was much lower than that in blood cell fraction DNA isolated from IV group. Additionally, no fragment was detected in the LM group. We used relatively smaller mice in our experiments, and only 240 μL of sample was used to extract cfDNA, resulting in a low yield of DNA. Therefore, it is likely that if the amount of plasma used to extract cfDNA is increased, the detection of transgene fragments could be more sensitive.

It has been reported that human candidate genes at a high risk of gene doping include *EPO*, insulin-like growth factor-1 (*IGF-1*), hypoxia inducible factor-1 (*HIF-1*), vascular endothelial growth factor-a (*VEGF-A*), and follistatin (*FST*), etc. [20,21,22]. Gene doping may enhance their expression. Moreover, silencing the gene expression of myostatin by RNA interference (RNAi) technology may lead to a risk of artificially increased muscle mass [23,24], since animal experiments have shown an increase in muscle mass using naked plasmid DNA expressed as a small hairpin RNA (shRNA) [23] or cholesterol-conjugated small interfering RNA (siRNA) [24]. Although there are many target genes that can be possibly used for gene doping, our PCR methods can detect only one decided transgene. Therefore, in near future, it is necessary to develop methods that can detect multiple different transgene fragments and different vectors simultaneously. In order to achieve this, a comprehensive analysis by next-generation sequencing (NGS) technology may be necessary.

Surprisingly, in ddPCR experiments, positive droplets were detected in DNA samples of rAdV-negative samples, although mice of the injected group and control group were completely separated. Additionally, ddPCR is believed to be highly precise and sensitive for the quantification of absolute values, according to Bio-Rad Laboratories, Inc. Therefore, there is a possibility of cross-contamination by transgenes of positive samples. Such cross-contamination must be absolutely avoided during the examination of gene doping. Therefore, we have the challenge to apply robot technology that resembles human moves for examination of gene doping. The robot named “Maholo”, developed by LabDroid (Robotic Biology Institute), demonstrates the concept that humanoid robots in laboratory can solve problems, such as numerous labor-intensive tasks required in high-throughput research, as well as the dangers (and costs) associated with experiments involving pathogens and harmful reagents [25]. Maholo can perform various tasks, including automating sample preparations in a clean room with a low risk of artificial contaminations. Currently, we have tested and established practicality and developed a computer system to apply testing for the detection of transgene fragments in our model mice by Maholo. This technology might be used in human applications in future.

## 5. Conclusions

In the present study, transgene fragments could directly be detected from blood cell fraction DNA, plasma cfDNA, and stool DNA by qPCR and ddPCR methods in a gene delivery mice model, depending on specimen type and injection methods. Additionally, it was observed that a combination of blood cell fraction DNA and ddPCR is more sensitive than other combinations used in this model. These results could accelerate the establishment of examination methods for gene doping.

## Figures and Tables

**Figure 1 genes-10-00436-f001:**
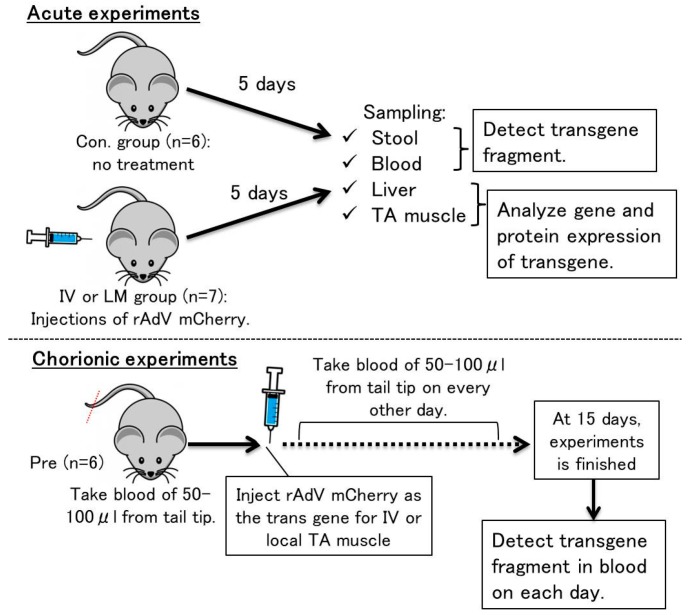
Overview of animal experiments carried out in this study.

**Figure 2 genes-10-00436-f002:**
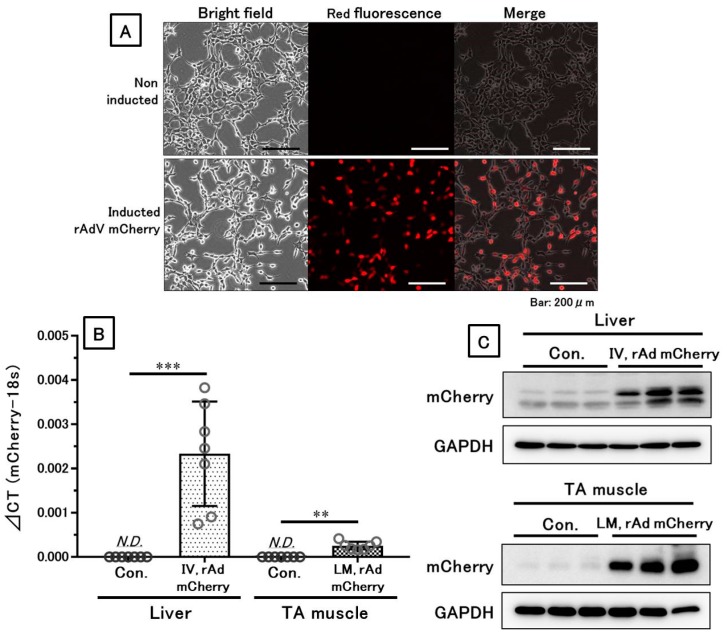
Confirmed gene and protein expression by the recombinant adenoviral (rAdV) vector. RNA and functional proteins were sufficiently expressed both in vitro and in vivo. (**A**) Functional protein expression of mCherry in HEK 293A cells. (**B**) Gene expression of *mCherry* in the liver or tibialis anterior (TA) muscle, detected by qPCR. (**C**) Protein expression of mCherry in the liver or TA muscle, shown by Western blotting of representative samples. IV means intravenous injection, and LM means local muscular injection of the rAdV vectors in the TA muscle. **: *p* < 0.01, ***: *p* < 0.001.

**Figure 3 genes-10-00436-f003:**
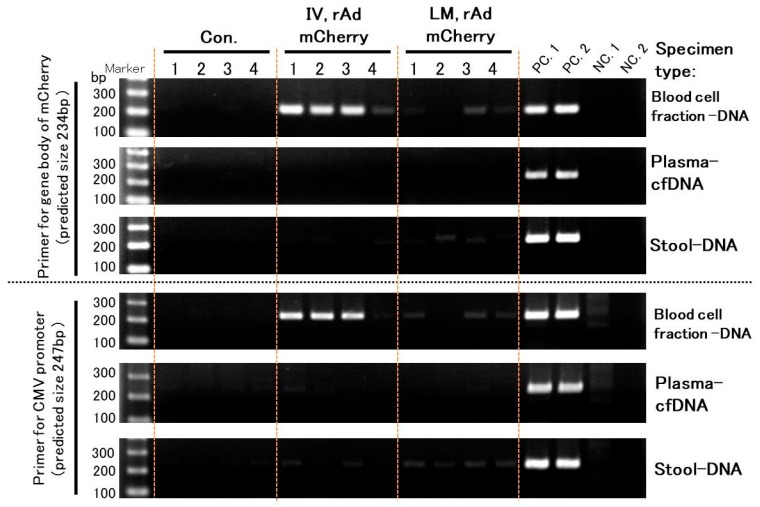
Detection of transgene fragments in representative samples using semi-quantitative PCR (sqPCR). Blood cell fraction DNA samples show strong signals, but other samples show very weak signals. IV means intravenous injection, and LM means local muscular injection of the rAdV vectors in the tibialis anterior (TA) muscle. PC. 1: positive control of rAdV DNA containing the *mCherry* gene (10 pg/µL). PC. 2: positive control of liver DNA containing the *mCherry* gene with a rAdV induction (10 ng/µL). NC. 1: negative control of mouse DNA (50 ng/µL). NC. 2: Distilled water (DW) as a negative control.

**Figure 4 genes-10-00436-f004:**
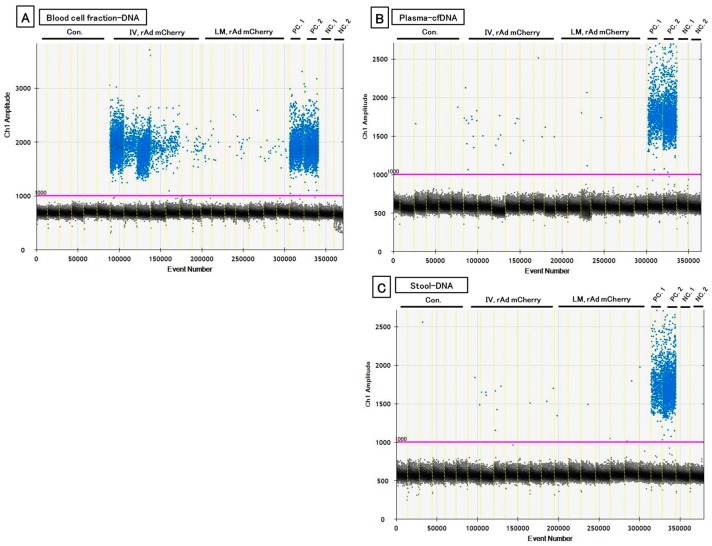
One-dimensional (1-D) plot data showing the detected transgene in each specimen by droplet digital PCR (ddPCR) reactions. (**A**) Blood cell fraction DNA. (**B**) Plasma plasma cell-free DNA (cfDNA). (**C**) Stool DNA. The blue plots denote positive and black ones denote the negative existence of transgene fragments. IV means intravenous injection, and LM means local muscular injection of the rAdV in the tibialis anterior (TA) muscle. PC. 1: positive control of rAdV DNA containing the *mCherry* gene (10 pg/μL). PC. 2: positive control of liver DNA with induced rAdV containing the *mCherry* gene (10 ng/μL). NC. 1: negative control of mouse DNA (50 ng/µL). NC. 2: distilled water (DW) as a negative control.

**Figure 5 genes-10-00436-f005:**
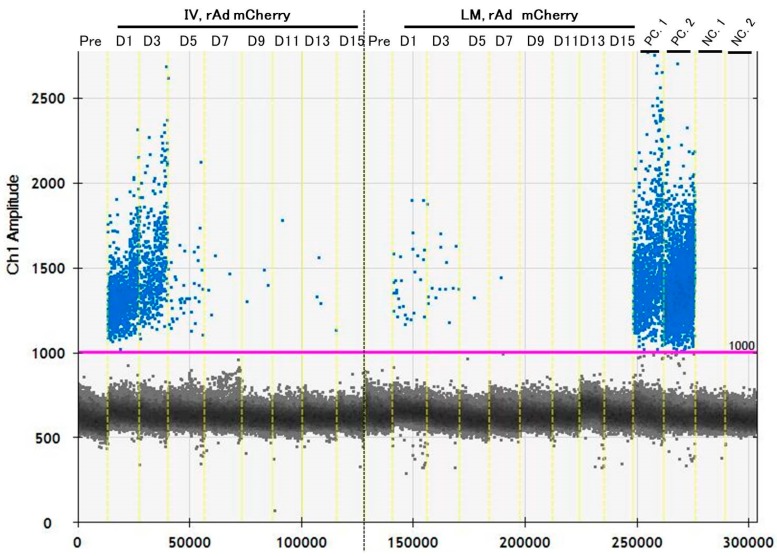
1-D plot of data showing detected transgene fragments in ddPCR reactions until 15 day after injection of the rAdV vectors. These plots show a mean positive for the transgenes in whole blood DNA pooled from six mice each day. Transgene fragments could be detected repeatedly. In particular, the detection of fragments was higher on days 1 (D1) and 2 (D2). The blue plots denote a positive presence and black plots denote a negative presence of transgene fragments. IV means intravenous injection, and LM means local muscular injection of the rAdV in the tibialis anterior (TA) muscle. PC. 1: positive control of rAdV DNA containing the *mCherry* gene (10 pg/µL). PC. 2: positive control of liver DNA containing the *mCherry* gene with induced rAdV (10 ng/µL). NC. 1: negative control of mouse DNA (50 ng/µL). NC. 2: distilled water (DW) as a negative control.

**Table 1 genes-10-00436-t001:** Types and relative numbers of the top seven clinically approved vectors used in gene therapy.

Vector	Gene Therapy Clinical Trials	
	Number	%
Adenovirus	541	18.5
Retrovirus	514	17.5
Naked/Plasmid DNA	452	15.4
Lentivirus	278	9.5
Adeno-associated virus	238	8.1
Vaccinia virus	133	4.5
Lipofection	119	4.1
Others	774	26.4
Total	2930	100

**Table 2 genes-10-00436-t002:** Primer sequences used in this study.

Methods	Targets	Sequences		Predicted Size (bp)
sqPCR	*mCherry* gene body	Forward	CACGAGTTCGAGATCGAGGG	234
		Reverse	GCCGTCCTCGAAGTTCATCA	
sqPCR	CMV promoter	Forward	CACGCCCATTGATGTACTGC	247
		Reverse	ACGCCAATAGGGACTTTCCA	
qPCR, ddPCR: Taq man probe assay	*mCherry* gene body	Forward	GGCACCAACTTCCCCTCC	115
		Probe	56FAM/CATGGTCTT/ZEN/CTTCTGCAT/3IABkFQ	
		Reverse	TCTGCTTGATCTCGCCCTTC	
qPCR: SYBR green assay	18s rRNA	Forward	AGTCCCTGCCCTTTGTACACA	70
		Reverse	CGATCCGAGGGCCTCACTA	

**Table 3 genes-10-00436-t003:** Detection of transgene fragments from qPCR. A high number of transgene fragments was detected in the blood cell fraction DNA in the IV group. IV means intravenous injection, and LM means local muscular injection of the rAdV in the tibialis anterior (TA) muscle. a: *p* < 0.01 vs. control (con.) within same specimens. b: *p* < 0.05 vs. local injection within same specimens. c: *p* < 0.05 vs. con. within same specimens.

Group	Mouse No.	Copy/μL of Transgene		
		Blood Cell Fraction-DNA	Plasma-cfDNA	Stool-DNA
**Con.**	1	0.0	0.0	0.0
	2	0.0	0.0	0.0
	3	0.0	0.0	0.0
	4	0.0	0.0	0.0
	5	0.0	0.0	0.0
	6	0.0	0.0	0.0
	**Median**	**0.0**	**0.0**	**0.0**
**IV, rAdV mCherry**	1	2528.2	33.4	3.7
	2	390.3	3.3	0.0
	3	1808.4	8.5	1.6
	4	217.9	5.1	0.0
	5	230.4	4.9	0.9
	6	26.2	5.5	0.8
	7	63.1	4.0	2.6
	**Median**	**230.4 ^a, b^**	**5.1 ^a, b^**	**0.9 ^a, b^**
**LM, rAdV mCherry**	1	56.2	0.0	1.8
	2	11.8	0.0	0.0
	3	16.5	0.0	0.0
	4	30.7	0.0	0.0
	5	15.9	0.0	0.0
	6	31.7	0.0	0.0
	7	28.3	0.0	0.0
	**Median**	**28.3 ^c^**	**0.0**	**0.0**

**Table 4 genes-10-00436-t004:** Detection of transgene fragments from droplet digital PCR ddPCR. Transgene fragments were detected in blood cell fraction DNA in the IV group. IV means intravenous injection, and LM means local muscular injection of the rAdV vector in the tibialis anterior (TA) muscle. a: *p* < 0.01 vs. con. within same specimens. b: *p* < 0.05 vs. local injection within same specimens. c: *p* < 0.05 vs. con. within same specimens. d: *p* = 0.089 vs. con. within the same specimens.

*Group*	Mouse No.	Copy/μL of Transgene		
		Blood Cell Fraction-DNA	Plasma-cfDNA	Stool-DNA
***Con.***	1	0.0	0.0	0.0
	2	0.0	0.0	0.0
	3	0.0	0.8	0.7
	4	0.0	0.0	0.0
	5	0.6	0.0	0.0
	6	0.0	0.8	0.0
	**Median**	**0.0**	**0.0**	**0.0**
***IV, rAdV mCherry***	1	4460.0	19.2	2.3
	2	620.7	0.7	2.2
	3	2873.3	6.0	5.5
	4	276.7	4.3	0.0
	5	190.0	2.3	0.0
	6	13.7	1.9	0.8
	7	42.7	2.5	1.5
	**Median**	**276.7 ^a, b^**	**2.5 ^a, b^**	**1.5 ^a^**
***LM, rAdV mCherry***	1	34.0	0.0	0.7
	2	5.5	0.0	0.0
	3	4.5	3.0	0.7
	4	12.9	1.4	0.0
	5	3.8	0.0	0.7
	6	16.5	0.0	1.4
	7	11.5	0.0	1.5
	**Median**	**11.5 ^c^**	**0.0**	**0.7 ^d^**

**Table 5 genes-10-00436-t005:** Repeated detection of transgene fragments using ddPCR.

	Mouse No.	Copy/μL of Transgene								
		Pre	1 day	3 days	5 days	7 days	9 days	11 days	13 days	15 days
**IV, rAdV *mCherry***	**1**	0.0	817.0	1398.0	48.0	4.3	1.4	0.0	0.8	0.0
	**2**	0.0	890.0	209.0	25.0	2.4	0.0	0.7	3.4	0.7
	**3**	0.8	1108.0	796.0	10.3	0.9	0.0	0.0	10.4	0.0
	**4**	2.0	1422.0	298.0	13.7	1.4	2.7	1.7	0.0	0.0
	**5**	0.7	1900.0	1132.0	62.0	10.5	1.4	1.6	1.5	0.8
	**6**	0.0	4800.0	1261.0	39.0	7.1	2.0	0.7	0.9	1.5
	**Median**	**0.4**	**1265.0**	**964.0**	**32.0**	**3.4**	**1.4**	**0.7**	**1.2**	**0.4**
	**p- values vs Pre**		**0.0002**	**0.0004**	**0.0063**	**0.0911**	**0.4181**	**0.5886**	**0.2992**	**0.6177**
**LM, rAdV *mCherry***	**1**	0.0	70.0	66.0	7.3	0.7	0.8	0.0	0.0	0.0
	**2**	0.9	149.0	24.0	5.0	1.6	0.7	0.0	0.0	0.0
	**3**	0.0	19.0	7.2	4.2	0.7	0.0	0.0	0.0	0.0
	**4**	0.0	10.4	4.6	0.8	0.8	0.7	0.0	0.0	0.0
	**5**	0.0	8.6	9.0	0.0	0.8	0.8	0.0	1.5	0.0
	**6**	0.0	5.2	27.0	4.1	0.0	0.7	0.0	0.0	0.0
	**Median**	**0.0**	**14.7**	**16.5**	**4.2**	**0.8**	**0.7**	**0.0**	**0.0**	**0.0**
	**p- values vs Pre**		**0.0012**	**0.0012**	**0.0666**	**0.2576**	**0.2902**	**0.5904**	**0.7753**	**0.5907**

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
