# Peer review of "Detection of Transgenes in Gene Delivery Model Mice by Adenoviral Vector Using ddPCR"

_genes, 2019, doi:10.3390/genes10060436_

Round 1
Reviewer 1 Report
The manuscript describes different PCR methods by which AdV vectors can be detected in tissue/bodiliy fluids. Although the data presented here can be very useful for further development of molecular AdV detection methods in vivo, major improvements to the manuscript are necessary. To further improve the manuscript comments are placed in each section (see attachment). As a general note, data presentation, the rationale for doing certain experiments, the number of experiments, certain controls, detaild description of the vector used but most importantly data interpretation needs a lot more work.

Author Response
Thank you for your comment. We answered as following. The revised sentences are shown as yellow highlight.
Response about English problems:
The original manuscript was corrected to correct English by Japanese proofreading company (https://www.scientific-language.co.jp/). However, The English calibration may have been inadequate. And then, we submit the revised manuscript. If there are any English problems even now, we would like to actively revise it. Therefor please tell us the proper way to calibrate English. As the one way to calibrate the English recommended form the journal editor, there is to request for MDPI author services (https://www.mdpi.com/authors/english)
Line number 70-71: language
Our answer
Sorry. We believed that this sentence have correct. Please tell us the appropriate text.
Line number 81: Reference? very strong claim
Our answer
This sentence is based on experience. Therefor the sentence was removed.
Line number 120: In my opinion this section does not add much value. It would suffice to add the information when each plasmid is mentioned in section 2.2
Our answer
The information of plasmids transferred to the next section.
Line number 128: Possibly start the section with AdVs were generated encoding a mCherry transgene in E1? There is no justification added here why this gene was selected so this is redundant information in this part.
Our answer
Yes. The mCherry gene was inserted in the region on E1. We removed the sentence.
Line number 128-129: reformulate. Reads as if the restriction sites are the template. I assume that the authors want to say that the primers or the amplified product was flanked by the restriction sites?
Our answer
The following corrections were made and reflected in the new manuscript at line number 128-130: “The mCherry gene having restriction enzyme sites of 5’-EcoRI and 3’-NotI was amplified by PCR with templated pcDNA3.1-Peredox-mCherry.”
Line number 131: Language
Our answer
We calibrated as following at the line number 130-132 of the revised manuscript: “Its gene was then cloned into pENTR4 plasmid between EcoRI and NotI sites by restriction enzyme digestion followed by ligation with T4 ligase (Promega).”
Line number 132-134: Is there a reference for this cloning method? What are the overhangs how long? Is the rAd vector checked by sequencing prior to rescue? Which vector is it? Ad5? or other? this information is necessary.
Our answer
The method of cloning by LR Clonase followed the manufacturer's protocol. The sequence was confirmed after ligation of mCherry gene to pENTR4 plasmid. pAd CMV DEST is used to create Ad5. Therefore we calibrated as following:
“The sequences of inserted mCherry gene in the pENTR4 plasmids were read using sanger sequencing and confirmed to be correct sequences.” at line number 132-133 of the revised manuscript.
“Using Gateway LR Clonase Enzyme mix (Thermo Fisher Scientific) and according to the manufacturer protocol, pENTR4 containing mCherry gene was allowed to react and recombine with pAd/CMV/V5-DEST (destination vector) in an LR reaction to move the mCherry gene into pAd/CMV/V5-DEST plasmid, which can make rAdV type 5 containing the trans-gene.” at line number 132-137 of the revised manuscript.
Line number 142: Reference? how about the infectious units? where those confirmed?
Our answer
We searched the reference and recalculated the virus concentration. In the first, we used the old method (Maizel J et al., 1968. Virology). However, Sweeney and Hennessey (2002) showed that the old method was wrong. Therefore, we recalculated and corrected the virus concentration and we calibrated the sentences as following:
“The concentration of rAdV viral particles (VP) was measured on a spectrophotometer, according to the method of Sweeney and Hennessey [23]” at the line number 144-146 of the revised manuscript. And then, the numbers of virus particles were calibrated at the line number 149 and 183 of the revised manuscript. We also added new reference article of number 23.
Line number 143: 10^5
Our answer
We calibrated at line number 147 of the revised manuscript.
Line number 145: 10^9
Our answer
We calibrated at line number 149 of the revised manuscript.
Line number 145: Was the linear range of detection mCherry determined?
Our answer
Yes we do. We determined the concentration at which the cells showed little damage and red fluorescence was visible enough.
Line number 196: as stated in the 2.3.1 section?
Our answer
Yes. We calibrated at line number 201 of the revised manuscript.
Line number 225:
Our answer
We calibrated at line number 230 of the revised manuscript.
Line number 275: Maybe add the rationale why its important to confirm the mCherry expression in both complimenting and non-complimenting cells.
Our answer
Our one purpose was to express functional mCherry proteins. To confirm the red fluorescence is important because these signals mean the expression of functional mCherry proteins. If cloning strategy was failure such as PCR error, functional protein don’t express in the cells. Therefore, our cloning strategy to make rAdV was success for expressions of the mRNA, protein and function in this study. We already described “To confirm the expression of functional mCherry protein----“ at line number 146 of the revised manuscript in the sections of 2.1.
Line number 281:Start with why and which PCR methods were tested.
Our answer
sqPCR is popular and inexpensive method in gene detection methods. sqPCR is used in genotyping of eukaryote and examinations of micro bacteria. Therefor we started test using sqPCR. But this method had unclear result.
Line number 279: Unclear
Our answer
Sorry. We believed that this sentence have clear. Please tell us the appropriate text.
Line number 309: Maybe this can be plotted in a graph? it is difficult to read the data like this maybe like 2B?
Our answer
It is difficult to plot the graph like 2B because there was large difference between maximum and minimum values in Table3.
Line number 318: Very technical graphs. The manuscript would benefit if this was plotted differently and not only showing the representative samples. Similar to 2B so that the statistics can be shown as well?
Our answer
The 1-D plot figure is characteristic figure on the ddPCR. The blue plots mean positive signals of the trans-gene fragments. This plot also visually provides amounts of the trance genes, which fully provide information of the transgene fragments as the visual. The table 4 data was converted from the 1-D plot data. Therefore, it can’t convert for graph like 2B.
We modified the Fig. 4 data. The Fig. 4 showed all sample data. Therefore, we removed “representative” at line number 322 of the revised manuscript.
Line number 318: Add rationale. Why was only the ddPCR used for chronic experiments?
Our answer
From the results of the acute experiments, combination method of ddPCR and blood cell fraction-DNA was highest sensitivity to detect the transgene fragments and it was hypothesized that performing ddPCR with whole blood-DNA is useful for to detect the transgene fragments for several days. Therefore, we performed the combination method in the chronic experiment.
Therefore we added and modified the sentence as above at line number 377-341 of the revised manuscript.
Line number 339: Again consider plotting the data differently for clarity.
Our answer
These plots mean positive of the transgene in whole blood-DNA pooled DNA of 6 mice each day. Therefore, we can’t show individual data on the 1-D plot data. But we already showed individual data at table 5. The table 1 data also has large variation. Therefor is difficult to show as a graph data of average value. And we modify the sentence as line number 347-348 in the revised manuscript.
Line number 362: Correlation analysis between the different methods? how many times was each detection performed?
Our answer
This detection was conducted as duplicate measurements and correlation analysis was not conducted. We think that there is positive and strong Correlation between qPCR and ddPCR.
Line number 391-397: Introduction?
We want to say that necessary to develop methods able to detect multiple different trans-gene fragments and different vectors simultaneously. Therefor we added the new sentence as line number 407-411 in the revised manuscript.
.
Line number 419: Is this necessary?
We think that this picture isn’t necessary. We removed the picture.
Best regards
Dr. Takehito sugasawa
Mr. Kai aoki

Reviewer 2 Report
A few comments:
- Results section. P11, l 355-358: While the droplet digital (ddPCR) permitted to detect a low amount of the transgene in the blood cell fraction-DNA via the LM introduction. What is the meaning of its absence in plasma-DNA and stool-DNA? Recombinant AdV is poorly spread or were there any technical problems?
- Discussion section
a) To meet the expectation of the authors, I fully disagree with the implementation of any doping system, because the commonly used substances are stimulants that can be detected in blood or muscles. As communicated by the authors in P.11 l.368-369, they are peptides rather than piece of DNA! Here the findings highlight the efficiency of the ddPCR that can give signals until 3 days post-injection.
b) Regardless of the injection way to introduce the recombinant AdV and to meet the objective of the study in expanding the timing of detection, should they improve the ddPCR or the viral vector?
c) P. 12, l. 401-417: It is crucial that any PCR-detection experiments were conducted in either a virus or a transgene-free area! I appreciate the correctness of the authors to point out the occurrence of false-positive through cross-contamination. However any scientific findings should be significantly objective so I would avoid any contradiction with the present findings, but I found inopportune to make some advertisement with the robot. The authors can shorten this item by simply claiming some requirement of any special equipment like a robot, avoiding any cross contamination.
Author Response
Thank you for your comment. We answered as following. The revised sentences are shown as yellow highlight.
1. Results section. P11, l 355-358: While the droplet digital (ddPCR) permitted to detect a low amount of the transgene in the blood cell fraction-DNA via the LM introduction. What is the meaning of its absence in plasma-DNA and stool-DNA? Recombinant AdV is poorly spread or were there any technical problems?
Our view:
Amounts of plasma-Cell-free DNA were very low, which can’t measure the amounts on nano-drop as spectrophotometer. Therefor there is possible that when the amounts of plasma-Cell-free DNA are increased, the positive signal of the transgene could be also increased.
In the blood cell faction-DNA, the positive signals were highest. Therefore it was considered that rAdV is captured in immune system cells as an immune response or penetrated into the red blood cells. Since this idea is only hypothesis, further consideration is needed in the future.
In stool-DNA, the positive signals were very low, even though the DNA amounts were enough. Therefor we considered that the amounts to leak trans-gene fragments for stools is very low or it is degraded by enteric bacteria. Since this idea is only hypothesis, further consideration is needed in the future.
2. Regardless of the injection way to introduce the recombinant AdV and to meet the objective of the study in expanding the timing of detection, should they improve the ddPCR or the viral vector?
Our view:
In this study, we used rAdV vectors. This vector can temporarily induce gene expression in liver. And then the rAdV vectors will disappear as time passes. Therefor the positive signals in ddPCR were also disappear as time passes. But, we think that when the extraction method is improved such as to concentrate transgene fragments like an immunoprecipitation, positive signals could be increased for long times.
3. P. 12, l. 401-417: It is crucial that any PCR-detection experiments were conducted in either a virus or a transgene-free area! I appreciate the correctness of the authors to point out the occurrence of false-positive through cross-contamination. However any scientific findings should be significantly objective so I would avoid any contradiction with the present findings, but I found inopportune to make some advertisement with the robot. The authors can shorten this item by simply claiming some requirement of any special equipment like a robot, avoiding any cross contamination.
Our view:
We agreed the review. Therefore, the sentence was shortened and the Fig. 6 of robot is removed.
Response about English problems:
The original manuscript was corrected to correct English by Japanese proofreading company (https://www.scientific-language.co.jp/). However, The English calibration may have been inadequate. And then, we submit the revised manuscript. If there are any English problems even now, we would like to actively revise it. Therefor please tell us the proper way to calibrate English. As the one way to calibrate the English recommended form the journal editor, there is to request for MDPI author services (https://www.mdpi.com/authors/english)
Best regards
Dr. Takehito Sugasawa
Mr. Kai Aoki
